# Mucosal Vaccination Against SARS-CoV-2 Using Human Probiotic *Bacillus subtilis* Spores as an Adjuvant Induces Potent Systemic and Mucosal Immunity

**DOI:** 10.3390/vaccines13070772

**Published:** 2025-07-21

**Authors:** Raul Ramos Pupo, Laura M. Reyes Diaz, Gisela M. Suarez Formigo, Yusnaby Borrego Gonzalez, Miriam Lastre Gonzalez, Danay Saavedra Hernandez, Tania Crombet Ramos, Belinda Sanchez Ramirez, Roberto Grau, Niels Hellings, Piet Stinissen, Oliver Perez, Jeroen F. J. Bogie

**Affiliations:** 1Instituto de Ciencias Básicas y Preclínicas “Victoria de Girón”, Universidad de Ciencias Médicas de La Habana, Havana 11300, Cuba; 2Department of Immunology and Infection, Biomedical Research Institute, Hasselt University, 3590 Diepenbeek, Belgium; raul.ramospupo@uhasselt.be (R.R.P.);; 3University MS Center, 3590 Diepenbeek, Belgium; 4Center of Molecular Immunology, Havana 11300, Cuba; dxs4415@med.miami.edu (D.S.H.);; 5Abu Dhabi Stem Cells Center, Abu Dhabi P.O. Box 4600, United Arab Emirates; 6College of Medicine and Health Sciences, United Arab Emirates University, Al Ain P.O. Box 15551, United Arab Emirates; 7Biotechnology Department, Kyojin S.A., Rosario 2000, Argentina

**Keywords:** COVID-19, mucosal vaccination, *Bacillus subtilis* spores, systemic and mucosal immunity

## Abstract

**Background/Objectives**: The ongoing evolution of SARS-CoV-2 has highlighted the limitations of parenteral vaccines in preventing viral transmission, largely due to their failure to elicit robust mucosal immunity. **Methods**: Here, we evaluated an intranasal (IN) vaccine formulation consisting of recombinant receptor-binding domain (RBD) adsorbed onto human probiotic *Bacillus subtilis* DG101 spores. **Results**: In BALB/c mice, IN spore-RBD immunization induced strong systemic and mucosal humoral responses, including elevated specific IgG, IgM, and IgA levels in serum, bronchoalveolar lavage fluid (BALF), nasal-associated lymphoid tissue (NALT), and saliva. It further promoted mucosal B cell and T cell memory, along with a Th1/Tc1-skewed T cell response, characterized by increased IFN-γ-expressing CD4^+^ and CD8^+^ T cells in the lungs. **Conclusions**: All in all, these findings highlight the potential of intranasal vaccines adjuvanted with probiotic *B. subtilis* spores in inducing sterilizing immunity and limiting SARS-CoV-2 transmission.

## 1. Introduction

The Acute Respiratory Syndrome Coronavirus 2 (SARS-CoV-2), the virus responsible for COVID-19, has presented significant challenges to global public health [1,2,3,4,5,6]. To date, over 180 vaccine candidates have entered clinical trials [7,8,9], with 14 vaccines having received validation from the World Health Organization (WHO) by December 2024 [10]. These licensed vaccines have dramatically slowed the pandemic’s pace [11,12,13]. However, despite the substantial reduction in disease severity, hospitalizations, and mortality, the virus’s transmissibility has persisted, allowing it to spread and give rise to several variants of concern (VOCs) capable of evading the immunity generated [14,15,16]. This has led to a prolonged battle between vaccine development and viral evolution [17,18,19].

While licensed parenteral SARS-CoV-2 vaccines, including mRNA vaccines, have shown high efficacy in preventing severe COVID-19, a major drawback is their limited ability to prevent virus transmission. In this context, they mainly elicit strong systemic IgG responses; however, respiratory viruses like coronaviruses spread through droplets and initially infect the upper respiratory tract, a site that systemic IgG does not sufficiently protect. As a result, these vaccines often fail to provide sterilizing immunity, allowing local viral replication in the respiratory mucosa, which can promote further transmission and potentially drive the emergence of resistant VOCs. In contrast, mucosal vaccines induce the production of specific secretory IgA (sIgA), which is expected to be more effective than IgG in neutralizing SARS-CoV-2. By blocking the virus at the portal of entry, mucosal vaccines are more likely to prevent initial viral replication and offer sterilizing immunity. Finally, mucosal immunization elicits protective tissue-resident CD4^+^ and CD8^+^ T cell responses within local lymphoid tissue. Despite the respiratory nature of SARS-CoV-2 infection and the potential benefits of mucosal immunization, only 10% of vaccine candidates are designed for mucosal delivery, with none yet approved [7,10,20,21,22,23,24].

In this study, we evaluated a novel nasal-delivered vaccine formulation and investigated its capacity to induce both systemic and mucosal adaptive immunity. To enhance its immunogenic potential, we utilized human probiotic *Bacillus subtilis* DG101 spores [25,26,27] as a novel adjuvant. *B. subtilis* is an aerobic, non-pathogenic Gram-positive soil bacterium, which is widely used as a GRAS (Generally Recognized as Safe by the FDA) food ingredient for humans (e.g., natto, a traditional Japanese food) and animals, and represents an important host for the production of medicinal proteins and industrial enzymes. Its spores have proven safe and effective as vaccine carriers and adjuvants for mucosal vaccines against pathogens [28,29,30,31,32]. Moreover, they have been reported to have intrinsic immunomodulation properties by stimulating both innate and adaptive immune responses [33,34,35]. The *B. subtilis* spores further exhibit heat-resistant properties that enhance vaccine stability and facilitate distribution [25,36]. These unique characteristics make *B. subtilis* DG101 spores a promising candidate for ensuring high safety, efficacy, and room-temperature stability in mucosal vaccines. The receptor-binding domain (RBD) was chosen as a target as it engages with the angiotensin-converting enzyme 2 (ACE2) on host cells, allowing SARS-CoV-2 to enter the cells. RBD is small (~200 aa and ~25 kDa) and folds autonomously, making it suitable for this platform [37,38,39]. In preclinical and clinical trials, recombinant RBD in multiple vaccine formulations has induced robust neutralizing immunity [40,41,42,43,44,45]. Our findings indicate that the intranasal co-administration of RBD and human probiotic *B. subtilis* DG101 spores generates potent specific humoral and cellular mucosal responses in the respiratory tract. We anticipate that these spores’ hydrophobic and electrostatic properties [46,47] readily allow the non-covalent binding of RBD on their surface, thus mimicking the natural viral infection. Overall, this mucosal vaccination strategy induces systemic responses comparable to parenteral vaccines while providing the crucial advantage of mucosal-specific immunity, making it a more effective approach to limiting virus transmission.

## 2. Materials and Methods

### 2.1. Animals

Female BALB/c mice, aged 8 to 12 weeks, were obtained from Envigo or CENPALAB (National Center for the Production of Laboratory Animals, Havana). Animals were maintained under specific pathogen-free conditions and managed in accordance with the principles and procedures established in Council Directive 86/609/EEC. Mice were housed on a 12 h light/dark cycle with unrestricted access to water and a standardized, grain-based maintenance diet formulated for laboratory rodents. The investigation was approved by the Hasselt University Ethical Committee for Animal Experimentation and the Institutional Animal Care and Use Committee of the Center of Molecular Immunology. To assess immune responses, blood samples were collected at days 35 (only for IgM represented in Figure 1C) and 42 (all the other assays), saliva at day 38, and bronchoalveolar lavage fluid (BALF) at day 42. Additional serum was collected on day 180 to evaluate long-term immunity (only represented in Figure 1E). Nasal-associated lymphoid tissue (NALT), spleen, and lungs were harvested for cellular analysis at day 42 (Figure 1A).

### 2.2. Vaccine Formulation and Inoculation

Recombinant SARS-CoV-2 Wuhan-Hu-1 (aa 328–533) RBD-His protein was produced in a HEK293 cell expression system [48]. A total concentration of 20 µg was used for intramuscular doses. For the intranasal inoculations, 80 µg was selected based on our preliminary dose-escalation experiments, where this amount induced the highest mucosal immune activation with our formulation. This can be explained by the capacity of *B. subtilis* spores to adsorb large amounts of antigen [38], particularly relevant for the small RBD protein (~25 kDa) [39], and the lower bioavailability typically associated with the intranasal route, where higher antigen quantities are often required to overcome mucociliary clearance and enhance delivery to immune-inductive sites [49,50]. Spores were obtained after culture of the human probiotic *B. subtilis* DG101 strain (ICBP, Havana) derived from the natto strain [25,51]. A total concentration of 1 × 10^9^ *B. subtilis* DG101 spores was used for all the inoculations. The vaccine was formulated in PBS (Gibco^TM^, Thermo Fisher Scientific, Waltham, MA, USA) by mixing the RBD and the spores, accordingly, and following an incubation step of 1 h at room temperature (RT) and 100 rpm shaking to allow the antigen adsorption [52,53,54,55,56]. For intramuscular delivery, the left quadriceps was injected with 50 µL using a 31G syringe (Micro-Fine^TM^, BD, Franklin Lakes, NJ, USA). For intranasal vaccination, each nostril was inoculated by pipetting 10 µL of the formulation using 10 µL tips (Sarstedt, Nümbrecht, Germany).

### 2.3. Blood Sampling

Mice were manually restrained, and the submandibular vein was accessed laterally to the mandible. A 25G needle (Henke Sass Wolf, Tuttlingen, Germany) was used to puncture the vein. Approximately 100 μL of blood was collected into a microcentrifuge tube (Greiner Bio-One, Kremsmünster, Austria) for further serum separation and downstream applications. Hemostasis was achieved by applying gentle pressure with sterile gauze for 30–60 s post-collection [57].

### 2.4. Saliva Collection

To stimulate salivation, mice were injected intraperitoneally (IP) with pilocarpine hydrochloride (Sigma-Aldrich, St. Louis, MO, USA) at 10 mg/kg body weight in 100 µL PBS (Gibco^TM^). Immediately after injection, mice were held slightly reclined to allow easy access to the oral cavity. Saliva was collected for 5 min following pilocarpine administration. Saliva droplets were gently aspirated from the oral cavity using a transfer pipette (Sarstedt) and stored in a microcentrifuge tube (Greiner Bio-One). Approximately 200 µL was recovered per mouse. Collected samples were placed on ice and diluted *v*/*v* in a 2× protease inhibitor cocktail solution (cOmplete ULTRA Tablet, Mini, Roche, Basel, Switzerland) to minimize IgA degradation. Animals were monitored throughout the procedure for signs of distress or adverse reactions to pilocarpine. After collection, mice were injected IP with 200 µL PBS (Gibco^TM^) to prevent dehydration, returned to their cages, and observed until complete recovery [58].

### 2.5. Transcardial Perfusion

Mice were IP injected with pentobarbital (Dolethal^R^, Vetoquinol, Lure, France; 200 mg/kg) before transcardial perfusion with a heparin sodium salt (Sigma-Aldrich) in PBS (Gibco^TM^) solution (20 U/mL). Once fully anesthetized, the thoracic cavity was opened to expose the heart. A 23G needle (Henke Sass Wolf) attached to the perfusion system was inserted into the heart’s left ventricle. The right atrium was immediately incised to allow exsanguination at a steady rate of 5 mL/min until the effluent ran clear. To further remove the blood from the lungs, the needle was pushed deeper into the right ventricle, and the left atrium was immediately incised, leading to complete blood removal [59].

### 2.6. Spleen Single-Cell Suspension

After perfusion, spleens were isolated and placed into 12-well plates (Greiner Bio-One) holding 1 mL of ice-cold RPMI 1640 medium (Gibco^TM^) per well. Tissues were mechanically dissociated using the plungers of 5 mL syringes (Henke Sass Wolf) by pressing them through 70 μm cell strainers (Greiner Bio-One) placed over 50 mL conical tubes (Greiner Bio-One). The resulting single-cell suspensions were centrifuged at 300× *g* for 10 min at 4 °C. Cell pellets were resuspended in 1 mL of ACK lysis buffer (Gibco^TM^) and incubated for 3 min at RT with gentle mixing to lyse red blood cells. This process was stopped with 10 mL of culture medium compounded by RPMI 1640 supplemented with 10% fetal bovine serum, 1% non-essential amino acids, 1% sodium pyruvate, and 1% penicillin/streptomycin (all from Gibco^TM^), followed by centrifugation at 300× *g* for 10 min. After discarding the supernatant, the resuspended cells in 2 mL of culture medium were filtered through a new 70 µm filter (Greiner Bio-One). The centrifugation step was repeated, and the cells were finally resuspended in 4 mL of culture medium. To assess viability, cells were counted using a Neubauer hemocytometer (Hausser Scientific, Horsham, PA, USA) after trypan-blue (Biochrom AG, Berlin, Germany) exclusion. Single-cell suspensions were used immediately for downstream applications [60].

### 2.7. Lung Single-Cell Suspension

The isolated lungs from the perfused mice were placed into 12-well plates (Greiner Bio-One) holding 1 mL of ice-cold RPMI 1640 medium (Gibco^TM^) per well. Next, the lungs were transferred into new plates and minced into small fragments using sterile scissors in 1 mL of digestion buffer per well. This buffer was compounded by culture medium, 1 mg/mL collagenase D, and 0.1 mg/mL DNase I (both from Roche). The tissue suspensions were incubated for 45 min at 37 °C, shaking at 15 min intervals to facilitate enzymatic digestion. Next, the resulting lung fragments were mechanically disrupted by pressing them using the plungers of 5 mL syringes (Henke Sass Wolf) through 70 μm cell strainers (Greiner Bio-One) placed over 50 mL conical tubes (Greiner Bio-One). Cells were washed through the strainer with an additional culture medium to maximize recovery, followed by centrifugation at 300× *g* for 10 min. After discarding the supernatant, 1 mL of 10 mM EDTA (Sigma-Aldrich) in PBS (Gibco^TM^) was added to resuspend the cells and stop the digestion, followed by 2 mL of culture medium. Red blood cell lysis and cell counting were performed as described previously for splenic cells [61].

### 2.8. BALF Collection

A small incision was made in the trachea, and a 20G catheter (Insyte^TM^, BD) was carefully inserted and secured with surgical thread (SMI). The lungs were lavaged by slowly instilling 500 µL of PBS (Gibco^TM^) into the lungs with a 1 mL syringe (Henke Sass Wolf), followed by gentle aspiration. The retrieved volume was transferred to a microcentrifuge tube (Greiner Bio-One), placed on ice, and diluted as described above for saliva to minimize immunoglobulin degradation [62].

### 2.9. NALT Isolation and Culture

Following perfusion, the upper palates were dissected by tracing along the inner edges of the incisors and molars. The palates were then carefully lifted using forceps to avoid tearing and transferred individually into 48-well plates (Greiner Bio-One) containing 250 μL of culture medium per well. After successive washing steps, NALTs were incubated at 37 °C in 5% CO_2_. After 24 h, 200 μL of supernatant medium from culture wells was collected in microcentrifuge tubes (Greiner Bio-One) for further analysis [63].

### 2.10. Specific Antibody Determinations

Anti-SARS-CoV-2 RBD-specific antibodies were evaluated by Enzyme-Linked Immunosorbent Assay (ELISA). The working solutions were the following: coating buffer at 0.1 M (pH 9.6) consisting of 3.18 g sodium carbonate (Na_2_CO_3_, Sigma-Aldrich) and 5.86 g of sodium bicarbonate (NaHCO_3_, Sigma-Aldrich) per liter of distilled water; blocking buffer consisting of Tween 20 (Sigma-Aldrich) 0.05% *v*/*v* and fat-free milk (Marvel, Premier Foods, St Albans, UK) 4% *m*/*v* in PBS (Gibco^TM^); assay buffer consisting of Tween 20 (Sigma-Aldrich) 0.05% *v*/*v* and fat-free milk (Marvel) 2% *m*/*v* in PBS (Gibco^TM^); and PBS-T prepared with Tween 20 0.05% *v*/*v* in PBS (Gibco^TM^). RBD was bound to the surface of flat-bottom 96-well microtiter plates (MaxiSorp, NUNC, Thermo Fisher Scientific) by adding 50 μL per well of coating buffer containing the antigen at 5 μg/mL. The plates were incubated overnight at 4 °C with 50 μL per well. Next, 150 μL of blocking solution was applied per well, and the plates were incubated for 30 min at 37 °C. Serum, BALF, saliva, or NALT culture supernatant was diluted (1:1000 for serum and 1:2 for the others) in assay buffer, and 50 μL per well was placed for 2 h at 37 °C. The washings were conducted 3 times using 250 µL of PBS-T per well. Subsequently, the corresponding conjugate was added for 1 h at 37 °C in a volume of 50 μL per well: goat anti-mouse IgM-HRP (Sigma-Aldrich), goat anti-mouse IgG-HRP (Sigma-Aldrich), goat anti-mouse IgG1-HRP (Sigma-Aldrich), or goat anti-mouse IgG2a-HRP (Sigma-Aldrich), all diluted 1:10,000 in assay buffer, or biotin rat anti-mouse IgA (BioLegend, San Diego, CA, USA) diluted 1:2000 in assay buffer. Afterward, the plates were washed again, similarly. For IgA, an extra incubation step was needed for adding 50 μL of streptavidin-HRP (BioLegend) diluted 1:2000 in assay solution, followed by another washing. For developing the colorimetric reaction, 100 μL of TMB substrate (Thermo Fisher Scientific) was added per well and incubated in the dark for 20 min at RT. Reactions were stopped with 50 µL of 1 M sulfuric acid (H_2_SO_4_, Sigma-Aldrich), and the absorbance was measured at 450 nm in a microplate reader (CLARIOStar Plus, BMG Labtech, Ortenberg, Germany). Titers were evaluated following 2-fold serial dilutions. The endpoint titer was demarcated as the highest serum dilution, yielding an absorbance at least four times greater than that of preimmune serum diluted 1:50 [64,65].

### 2.11. Avidity Index

Antibody–antigen binding strength was evaluated by measuring the resistance to disruption with the chaotropic agent urea. Identical ELISA plates to those above were coated and blocked, as formerly described, for specific IgG antibody determinations. The sera were added in the minimal dilution so that the absorbance value exceeded 1 in the titration ELISA, and incubated for 1 h at 37 °C. Afterward, a three-time washing with PBS-T was conducted. One set of wells was then washed an additional three times with PBS-T (without urea), while the other set was washed three times with 6 M urea (CH_4_N_2_O, Sigma-Aldrich) in PBS-T (urea-treated) to disrupt low-affinity antibody–antigen interactions. The following steps for the ELISA were as previously presented. The avidity index (AI) was calculated as (OD with urea/OD without urea) × 100, representing the proportion of high-affinity IgG that remained bound. Antibodies were classified as high-avidity if the AI exceeded 50% [65,66].

### 2.12. Neutralization Assay

A molecular virus neutralization assay was conducted to evaluate the capacity of antibodies to inhibit RBD binding to its corresponding cellular receptor, ACE2. RBD fused to human IgG1 Fc region (RBD-hFc) and ACE2 fused to murine IgG2a Fc region (ACE2-mFc) proteins were produced in a HEK293 cell expression system. Briefly, identical ELISA plates to those above were coated with ACE2-mFc 5 μg/mL in coating buffer. Then, 2-fold serial dilutions of sera in assay buffer were mixed 1:1 (*v*/*v*) with 40 ng/mL RBD-hFc and incubated for 1 h at 37 °C. Next, 50 µL of the mixtures were added to the wells and incubated for 2 h at 37 °C. The following steps for the ELISA were as described previously, but using the goat anti-human IgG-HRP (Sigma-Aldrich) diluted 1:10,000 in assay buffer. Maximal recognition corresponds to RBD-hFc (40 ng/mL) mixed 1:1 (*v*/*v*) with assay buffer. The molecular viral neutralization titer (mVNT_50_) was defined as the highest serum dilution, resulting in 50% inhibition of maximal recognition [67].

### 2.13. CFSE-Proliferation Assay

Lung single-cell suspensions were cultured at 1 × 10^6^/well in 96-well plates (Greiner Bio-One) and antigen recall was conducted by adding RBD at 5 μg/mL. CFSE Cell Division Tracker Kit (BioLegend) was used according to the manufacturer’s protocol. Surface staining (B and T cells) was performed, followed by fixation and permeabilization, and subsequent intracellular staining (T cells) as described below for flow cytometry [68].

### 2.14. Flow Cytometry

The following reagents and fluorophores were used: Alexa Fluor^®^ 700 anti-mouse/human CD45R/B220 (1:200, clone RA3-6B2, BioLegend), Alexa Fluor^®^ 700 anti-mouse CD45 (1:200, clone 30-F11, BioLegend), Biotin anti-mouse IgA (1:200, clone RMA-1, BioLegend), APC Streptavidin (1:200, BioLegend), APC anti-mouse CD103 (1:200, clone 2E7, BioLegend), Brilliant Violet 421^TM^ anti-mouse CD138 (1:100, clone 281-2, BioLegend), Brilliant Violet 510^TM^ anti-mouse CD8a (1:200, clone 53-6.7, BioLegend), Brilliant Violet 605^TM^ anti-mouse IgD (1:200, clone 11-26c.2a, BioLegend), Brilliant Violet 650^TM^ anti-mouse CD19 (1:200, clone 6D5, BioLegend), Brilliant Violet 650^TM^ anti-mouse/human CD44 (1:200, clone IM7, BioLegend), Brilliant Violet 785^TM^ anti-mouse CD69 (1:200, clone H1.2F3, BioLegend), FITC anti-mouse CD3 (1:500, clone 17A2, BioLegend), Pacific Blue^TM^ anti-mouse CD4 (1:200, clone RM4-5, BioLegend), PE anti-mouse CD38 (1:200, clone 90, BioLegend), PE anti-mouse IL-4 (1:200, clone 11B11, BioLegend), PE/Cyanine7 anti-mouse IFN-γ (1:200, clone XMG1.2, BioLegend), PE/Cyanine7 Goat anti-mouse IgG (1:200, clone Poly4053, BioLegend), PE/Dazzle^TM^ 594 anti-mouse IL-17A (1:50, clone TC11-18H10.1, BioLegend), PerCP/Cyanine5.5 anti-mouse CD3 (1:200, clone 17A2, BioLegend). CFSE Cell Division Tracker Kit (BioLegend) and Zombie NIR^TM^ Fixable Viability Kit (BioLegend). RBD was conjugated to FITC (Appendix A). For stimulation and accumulation of intracellular cytokines, cells were incubated with phorbol 12-myristate 13-acetate (20 ng/mL; Sigma-Aldrich), calcium ionomycin (1 µg/mL; Sigma-Aldrich), and GolgiPlug^TM^ (2 µg/mL; BD Biosciences, San Jose, CA, USA) for 4 h before staining. Briefly, live/dead staining was conducted according to the manufacturer’s instructions for the Zombie NIR^TM^ Fixable Viability Kit (BioLegend). Next, extracellular staining was performed in 100 µL FACS buffer containing the appropriate stains with incubation for 15 min in the dark at RT, followed by a washing step. Cells were then fixed and permeabilized by means of the eBioscience^TM^ Foxp3/Transcription Factor Staining Buffer Set (Thermo Fisher Scientific), following the manufacturer’s instructions, and intracellularly stained for assessing IFN-γ, IL-4, and IL-17A. Samples were acquired using a BD LSRFortessa (BD Biosciences) flow cytometer. Compensation was performed using single-stained controls, and the analysis was conducted using FlowJo software v10 [69].

### 2.15. Experimental Design and Statistical Analyses

All statistical analyses were performed with the software GraphPad Prism^TM^ v10 and conveyed as mean  ±  SD. Data normality was confirmed, and, according to the number of groups in the data sets, two-tailed unpaired Student t-test or one-way ANOVA (post hoc: Tukey) was applied. Significance levels legend: * *p*  <  0.05, ** *p*  <  0.01; *** *p*  <  0.001 or **** *p*  <  0.0001.

## 3. Results

### 3.1. SARS-CoV-2 RBD Intranasal Immunization Adjuvanted with Human Probiotic Bacillus subtilis Spores Induces Specific Systemic and Mucosal Responses

To evaluate the potential efficacy of mucosal delivery of the *B. subtilis* spore-RBD vaccine formula in eliciting an Ig response, BALB/c mice were immunized intranasally (IN) with three doses of RBD (80 µg) adsorbed onto 1 × 10^9^ *B. subtilis* DG101 spores (spore-RBD) (experimental design depicted in Figure 1A). Our findings indicate that this approach induced a robust RBD-specific serum IgG response, significantly higher than that achieved with the gold-standard intramuscular (IM) regimen of three doses of RBD (20 µg) adsorbed onto aluminum hydroxide gel (alum-RBD) (Figure 1B). Three IM doses of the spore-RBD vaccine also elicited an elevated specific IgG response, albeit lower than that observed with both the IM alum-RBD and IN spore-RBD formulations (Figure 1B). A similar trend was observed for specific IgM levels in serum (Figure 1C). Consistent with the established ability of mucosal vaccines to enhance specific IgA serum levels, only IN immunization with the spore-RBD formulation generated a significant anti-RBD response of this class in serum (Figure 1D). To further quantify the immune response, systemic anti-RBD IgG titers were assessed across the adjuvanted groups, revealing high titers in all three groups, with the highest observed in mice receiving the IN spore-RBD formulation. Moreover, we found specific IgG titers persisting over 10^4^ at 180 days for most mice in the IN spore-RBD group (Figure 1E). Finally, affinity assays demonstrated enhanced IgG affinity maturation following IN spore-RBD immunization compared to the IM alum-RBD formulation (Figure 1F). Supporting this, the anti-RBD antibodies induced by the IN spore-RBD vaccine exhibited superior neutralization capacity, most effectively inhibiting the RBD−ACE2 interaction (Figure 1G). Collectively, these findings demonstrate that IN delivery of the spore-RBD vaccine induces a potent and broad humoral immune response, characterized by elevated serum specific IgG, IgM, and IgA levels.

Having established a potent systemic Ig response, we next evaluated respiratory mucosal immunity following IN or IM inoculation with the spore-RBD formulation. Only mice immunized with three doses of IN spore-RBD developed high levels of anti-RBD IgA and IgG in the supernatant of nasal-associated lymphoid tissue (NALT) cultures and bronchoalveolar lavage fluid (BALF) (Figure 1H–K). Similarly, only IN spore-RBD immunization increased anti-RBD IgA levels in saliva (Figure 1L). Neither the IN RBD (without adjuvant) nor IM spore-RBD induced a notable IgG and IgA response in BALF, NALT cultures, and saliva (Figure 1H–L). Finally, by using FITC-conjugated RBD, we found that IN spore-RBD immunization results in an increased presence of antigen-specific B cells (B220^+^CD19^+^RBD^+^) within lung tissue (Figure 1M,N), as compared to IM spore-RBD inoculation. These findings demonstrate that IN spore-RBD inoculation induces a robust, mucosal, and specific humoral immune response.

**Figure 1 vaccines-13-00772-f001:**
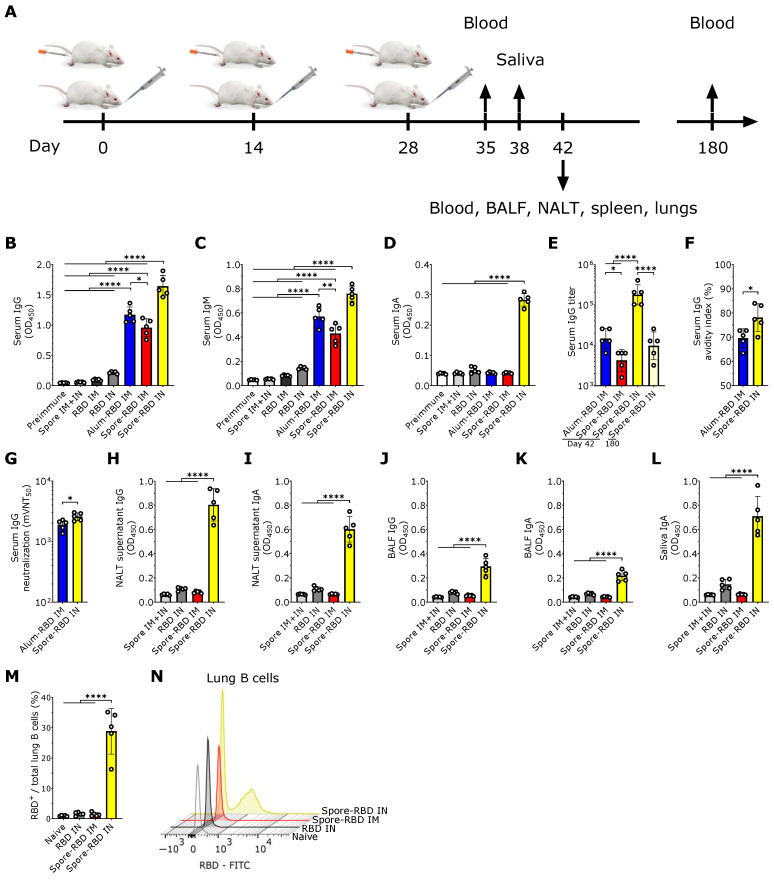
SARS-CoV-2 RBD intranasal immunization adjuvanted with *Bacillus subtilis* DG101 spores induces systemic and mucosal humoral responses. (**A**) BALB/c mice were immunized intranasally (IN) or intramuscularly (IM) with three doses of 80 µg or 20 µg, respectively, of recombinant SARS-CoV-2 Wuhan-Hu-1 receptor-binding domain (RBD) adsorbed onto 1 × 10^9^ *B. subtilis* DG101 spores at 14-day intervals. To assess immune responses, blood samples were collected on days 35, 42, and 180; saliva on day 38; and bronchoalveolar lavage fluid (BALF) on day 42. Nasal-associated lymphoid tissue (NALT), spleen, and lungs were harvested for cellular analysis on day 42. (**B**–**D**) Anti-RBD-specific antibodies in serum (dilution 1:1000): IgG on day 42 (**B**), IgM on day 35 (**C**), and IgA on day 42 (**D**). Groups include preimmune mice; mice immunized with spores alone IM and IN (spore IM + IN); RBD alone IM (RBD IM) or IN (RBD IN); RBD adsorbed onto aluminum hydroxide IM (alum-RBD IM); and RBD adsorbed onto spores IM (spore-RBD IM) or IN (spore-RBD IN). (**E**) Anti-RBD specific IgG titers after serial dilutions of the 42-day serum from alum-RBD IM, spore-RBD IM, and spore-RBD IN, and the 180-day serum from spore-RBD IN. (**F**) Serum IgG avidity index on day 42 after antigen–antibody dissociation using a chaotropic agent. (**G**) Half-maximal molecular virus neutralization titer (mVNT_50_), indicating the 42-day serum IgG dilution required to inhibit 50% of RBD–ACE2 binding. (**H**–**L**) Anti-RBD-specific mucosal antibodies (dilution 1:2): IgG and IgA on day 42 in NALT culture supernatants (**H**,**I**); IgG and IgA in BALF on day 42 (**J**,**K**); and IgA in saliva collected on day 38 (**L**). Groups include spore IM + IN, RBD IN, spore-RBD IM, and spore-RBD IN. (**M**,**N**) Percentage of anti-RBD-specific subset from the total lung B cells of naïve, RBD IN, spore-RBD IM, and spore-RBD IN (**M**). Representative fluorescence intensity histogram from one individual mouse per group, in panel M (**N**). All data are depicted as mean ± SD. Each experimental group consists of five animals, and each dot represents an individual animal. Statistical significance was calculated by one-way analysis of variance (ANOVA) for (**B**–**E**) and (**H**–**M**) or by Student’s t test for (**F**,**G**); * *p*  <  0.05, ** *p*  <  0.01; or **** *p*  <  0.0001.

### 3.2. SARS-CoV-2 RBD Intranasal Immunization Adjuvanted with Human Probiotic Bacillus subtilis Spores Induces Mucosal B Cell Memory in the Lungs

Essential immune effectors in long-term protection against SARS-CoV-2 in the lungs are resident memory B cells (B_RM_), which can rapidly differentiate in situ into antibody-secreting cells (ASCs) upon secondary challenge with cognate antigens [70,71]. Hence, we next assessed both subpopulations in the lungs of perfused mice, thereby circumventing carryover from blood lymphocytes. Our findings indicate that class-switched CD19^+/−^CD138^+^ASCs in lung tissue expressing IgA or IgG were significantly increased exclusively after IN spore-RBD inoculation (Figure 2A–D). A similar increase was observed for IgA^+^ or IgG^+^ class-switched B_RM_ (CD19^+^B220^+^IgD^-^CD38^+^) upon inoculation with spore-RBD (Figure 2E–H). To provide further evidence of efficient mucosal humoral memory, CFSE-labeled lung cells from immunized mice were analyzed for proliferative responses following antigen recall. CD3^-^B220^+^CD19^+^ B cells proliferated efficiently (Figure 2I,J), and the IgA^+^ subset expanded significantly upon RBD challenge only for IN spore-RBD (Figure 2K,L). These findings show that IN spore-RBD inoculation induces mucosal B cell memory in the lungs, resulting from both the overall expansion and an actual increase in percentage of the class-switched antigen-specific IgA^+^ and IgG^+^ subpopulations.

### 3.3. SARS-CoV-2 RBD Intranasal Immunization Adjuvanted with Human Probiotic Bacillus subtilis Spores Induces Systemic and Mucosal T Cell Immunity with a Th1 Bias

Given that IN vaccinations can elicit tissue-resident T cell responses within local lymphoid tissue [22], we next sought to characterize the CD4^+^ and CD8^+^ T cell response to spore-RBD inoculation. To this end, we immunophenotyped splenic and alveolar CD44^hi^CD4^+^ and CD44^hi^CD8^+^ T cells by flow cytometry to assess their production of IFN-γ, IL-4, and IL-17A, which are secreted from Th1/Tc1, Th2/Tc2, and Th17/Tc17 subsets, respectively. In the spleen, we found that mice immunized with spore-RBD either via IM or IN routes showed an increased percentage of IFN-γ-secreting CD4^+^ and CD8^+^ T cells, with IN immunization eliciting the most robust response (Figure 3A,B). CD4^+^ and CD8^+^ T cells expressing IL-17A and IL-4 were also increased in the IN and IM spore-RBD groups, though to a lesser extent, with no significant differences between the administration routes (Figure 3C–F). In the lungs, only IN immunization with spore-RBD increased IFN-γ and IL-17A-expressing CD44^hi^CD4^+^ and CD44^hi^CD8^+^ T cells (Figure 3G–J), with a more pronounced response observed in IFN-γ-expressing cells. No changes in IL-4 expression in CD4^+^ and CD8^+^ T cells were observed in the lungs following administration of spore-RBD, neither via IN nor IM routes (Figure 3K,L). These patterns are consistent with Th1 and Tc1 systemic and mucosal polarization predominance after IN spore-RBD. Supporting this premise, the distribution of IgG isotypes in the sera showed that IN immunization with spore-RBD induced higher levels of IgG2a relative to IgG1, compared to IM immunization (with spore- or alum-RBD), indicating a more Th1-skewed response upon IN inoculation. It is important to notice that the IgG2a/IgG1 ratio remained below 1, still signifying a slight Th2 bias (Figure 3M).

### 3.4. SARS-CoV-2 RBD Intranasal Immunization Adjuvanted with Human Probiotic Bacillus subtilis Spores Induces Mucosal T Cell Memory in the Lungs

In light of the mucosal humoral memory responses generated in the respiratory tract, we also considered assessing lung tissue-resident memory T cell (T_RM_) induction. Flow cytometry analysis revealed an enhanced proliferative response of CFSE-labeled CD4^+^ and CD8^+^ T cells from spore-RBD-immunized mice upon RBD recall (Figure 4A–D). Mirroring these findings, lung tissue from mice immunized with IN spore-RBD showed an increased presence of CD4^+^CD44^hi^CD69^+^CD103^+/−^ and CD8^+^CD44^hi^CD69^+^CD103^+/−^ T cells, indicative of enhanced T_RM_ formation (Figure 4E–J). Interestingly, we found no differences among the ratios CD103^+^/CD103^−^ for CD4^+^ T_RM_, which aligns with a proportional expansion of both subsets (Appendix A). However, for CD8^+^ T_RM_, the highest ratio was observed for the spore-RBD intranasally immunized mice, indicating a positive bias of the CD103^+^ subpopulation (Appendix A). Our results show that, in addition to the humoral mucosal responses, the IN spore-RBD vaccination strongly induces both CD103^+^ and CD103^-^ subpopulations of CD4^+^ T_RM_ and CD8^+^ T_RM_ in the lung parenchyma, in contrast to the IM formulation.

## 4. Discussion

As SARS-CoV-2 transmissibility and capability for immune evasion rebound with each VOC, developing novel vaccines that allow extensive protection against current and prospective VOCs becomes increasingly necessary. In this regard, mucosal vaccine delivery has garnered significant attention for its potential to induce broad protective immunity at the primary site of infection, thereby significantly reducing viral transmission [8,20,21]. Here, we present a preclinical evaluation of a novel IN mucosal vaccine formulation for COVID-19, composed of the RBD and human probiotic *B. subtilis* spores as an adjuvant. Spore-RBD inoculation elicited robust mucosal and parenteral immunity in a three-dose IN schedule, characterized by the expansion of antigen-specific CD8^+^ T_RM_, CD4^+^ T_RM_, B_RM_, and secretion of antigen-specific IgG and IgA. It also resulted in long-lasting immunity, with elevated specific IgG titers persisting after 180 days, indicating the presence of sustained, though waning, antibody responses. In contrast, IM immunization with this formulation and the gold-standard alum-RBD generated a modest systemic response without the desired mucosal immunity. These results indicate that human probiotic *B. subtilis* spores serve as a potent mucosal vaccine adjuvant and that their co-administration with the SARS-CoV-2 RBD may provide effective protection against COVID-19.

Both systemic and mucosal humoral immune responses are crucial against SARS-CoV-2. Concerning the former, serum-neutralizing antibodies targeting the RBD or the entire spike protein are key correlates of protection against symptomatic and severe COVID-19 [72,73,74,75], and are often used to assess vaccine efficacy [76,77,78]. However, systemic immunity alone cannot control infection at mucosal entry points, where SARS-CoV-2 invades through the nasal and oral passages, leading to respiratory tract infection and transmission. Current parenteral COVID-19 vaccines generate strong systemic but weak or no mucosal immunity, limiting their effectiveness in preventing breakthrough infections and spread [20,21,79].

Interestingly, we found elevated levels of specific IgM in serum at day 35 of the immunization timeline, corresponding to 7 days after the final dose. This timing supports a detectable response, particularly when induced through a mucosal tissue where immune dynamics differ from systemic responses. Specifically, intranasal delivery using *B. subtilis* spores may contribute to extended antigen retention, potentially maintaining an ongoing immune activation [49,50,52]. Consequently, the continued stimulation of naive B cells or the recall to expansion of IgM^+^ memory B cells could lead to our findings.

We show that IN spore-RBD induces a stronger systemic antibody response than IM. While some studies have reported higher systemic antibody responses with IM immunization, such differences likely depend on factors like adjuvant type, antigen delivery efficiency, and threshold concentration requirements for each route. For example, Yahyaei et al. (2025) found that twice the antigen dose was needed for IN delivery to match the systemic IgG levels achieved by IM administration of an mRNA-based influenza A vaccine [80], whereas Anthi et al. (2025) showed that IN vaccination with an RBD-based subunit vaccine produced higher systemic IgG than IM [81]. Hassan et al. (2021) similarly reported that an IN-administered spike protein chimpanzee adenovirus-vectored vaccine induced a superior systemic response than the IM route [82]. Our results align with these observations and can be explained by the potent adjuvant effect of *B. subtilis* spores and their capacity to deliver the antigen efficiently to mucosal sites, mimicking natural infection. We further show that IN spore-RBD inoculation results in a higher IgG2a/IgG1 ratio, suggesting a Th1-skewed response. Several factors could explain this outcome. The IN route engages the NALT, which provides a unique dendritic cell-rich microenvironment that promotes Th1 polarization [83]. *B. subtilis* spores could further enhance this effect by acting as a potent adjuvant and carrier, mimicking natural infection and delivering pathogen-associated molecular patterns (PAMPs) that activate pattern recognition receptors (PRRs) on dendritic cells, triggering Th1-skewing pathways [84]. All in all, the particulate nature of the spore-antigen complex may enhance antigen uptake and prolong presentation at mucosal surfaces, which would allow more efficient activation of antigen-presenting cells (APC) and differentiation of CD4^+^ T cells toward Th1 effectors [49,50].

IgA has been reported to govern the early SARS-CoV-2–specific antibody response in COVID-19 patients, both in serum and respiratory mucosa [85,86,87]. In 2024, Wagstaffe et al. highlighted that early mucosal IgA response is critical for viral control [23]. Our findings show that IN spore-RBD inoculation induces robust systemic and mucosal immunity, with high neutralizing titers and increased avidity of serum IgG, as well as serum and respiratory mucosal IgA. These findings position the spore-RBD vaccine formulation as a highly effective and promising method to attenuate both respiratory tract infection and transmission of SARS-CoV-2. We propose that after the IN immunization, locally activated and expanded B cell clones migrate to distant mucosal lymphoid tissues and the spleen, hence allowing humoral response in the upper and lower respiratory tract, and systemic immunity.

IgA in mucosal tissues is mostly dimeric and locally produced, attached to a secretory fragment (sIgA). Multivalency increases sIgA neutralization capacity. Accordingly, it is strongly associated with early mucosal SARS-CoV-2 control [23]. Whereas impaired mucosal IgA response in patients with severe COVID-19 has been reported, it is expected to result from plasmatic monomeric IgA (mIgA) reaching the airways through transudation, as found in BALF from severely ill patients. mIgA can trigger NETosis dysregulation, ultimately leading to severe autoinflammatory lung disease [88,89]. In contrast, sIgA is not involved in such a deleterious process, as steric hindrance prevents its interaction with Fcα receptors on neutrophils [89,90]. Importantly, we observed no major lung pathology based on macroscopic examination for IN spore-RBD, suggesting that the detected total IgA in respiratory secretions primarily represents sIgA in our experiments. Further research should provide more insight into the IgA response induced by our vaccine formulation and assess its safety and protective capacity upon viral challenge, in an appropriate animal model.

Analogous to circulating anti-SARS-CoV-2 antibodies, specific IgG and IgA in the respiratory mucosa are reported to diminish a few months after infection [85,87,91,92]. Therefore, an induction of mucosal memory B cells ensures a swift rise in local antibody levels upon re-exposure to the antigen, promoting sustained viral clearance and preventing further transmission in the long term [93,94,95]. Following a natural infection, RBD-specific B_RM_ were found in human lungs, emphasizing the importance of mucosal priming in the induction of these cells [96]. Our results indicate that lung tissue from mice challenged with IN spore-RBD was significantly enriched in both IgG^+^ and IgA^+^ memory B cells, which effectively proliferated after in vitro antigen recall. We therefore consider that the IN codelivery of the immunogen with this mucosal adjuvant might confer a protective mucosal and systemic immunity that efficiently reduces infection and transmissibility.

Antiviral immunity relies on the involvement and cooperation of T cells. In this regard, an efficient Th1 response is essential for orchestrating an effective immune defense against SARS-CoV-2, while cytotoxic T cells play a critical role in eliminating infected cells [93,97,98,99]. In contrast, Th2 bias after immunization has been linked to vaccine-associated enhanced respiratory disease [99,100,101]. Our results demonstrate a robust T cell response following IN spore-RBD inoculation, marked by a Th1/Tc1-biased immune profile without relevant Th2/Tc2 expansion, as evidenced by elevated IFN-γ without IL-4 responses in CD4^+^ and CD8^+^ T cells and a relatively higher IgG2a/IgG1 ratio. We also found induction of antigen-specific CD4^+^ and CD8^+^ T_RM_ in the lungs after IN spore-RBD inoculation. T_RM_ are the most abundant memory T cell subset in barrier tissues, which can proliferate locally upon antigen re-encounter and be primed to provide rapid, frontline immunity, playing a crucial role in long-lasting protection [102,103,104]. These findings underscore the importance of mucosal vaccines that directly target the airways, such as the spore-RBD formulation used in this study, positioning them as superior to peripheral vaccination strategies, which have failed to generate lasting lung T cell mucosal immunity against SARS-CoV-2 [93,105].

In this study, we not only developed a specific vaccine formulation for SARS-CoV-2 but also proposed an optimized platform for mucosal immunization based on human probiotic *B. subtilis* spores co-administered with any immunogen. This platform is expected to have broad applications, including booster immunization against novel SARS-CoV-2 variants in previously vaccinated or infected individuals, as well as primary immunization for unexposed individuals or against other emerging respiratory pathogens. Additionally, incorporating the RBDs of newly circulating SARS-CoV-2 variants into the formulation would allow the vaccine to evolve in parallel with the virus, and potentially curb its spread. At the same time, including distinct epitopes could enable a multivalent design capable of simultaneously targeting multiple strains or variants. Notably, *B. subtilis* spores are particularly attractive as a vaccine delivery system due to their stability, resistance to harsh environments, ease of handling, and ability to interact with immune cells [106,107,108,109,110]. Moreover, they are a valuable alternative to conventional mucosal adjuvants such as cholera toxin B subunit (CTB), heat-labile toxin (HLT) derivatives, and synthetic adjuvants (e.g., CpG-ODNs), which often raise toxicity and safety concerns [111]. Importantly, DG101 is a food-grade *B. subtilis* derived from the natto strain and widely characterized for its human probiotic properties, further supporting its excellent safety profile [25]. In particular, they are heat-stable, eliminating cold-chain requirements and enabling easy formulation, self-application, and thus broad global distribution. Additionally, their particulate nature facilitates efficient uptake by NALT, while their natural PAMPs, including peptidoglycan and lipoteichoic acid, provide strong intrinsic adjuvant effects through TLR2 and NOD2 activation [112]. This dual role as antigen carrier and innate immune activator enhances antigen uptake, presentation, and local immune activation. Although various *B. subtilis* strains are extensively documented as safe probiotics, they have not been employed as vaccine adjuvants in humans, with most prior studies relying on ones lacking human relevance, thereby limiting translational potential [106,113,114,115]. All in all, by using spores from a human probiotic strain, our findings point to a safe, economical, and effective new vaccination strategy capable of inducing both systemic and mucosal immunity.

While our findings are promising, this study has some limitations that should be acknowledged. Protective efficacy was not evaluated through viral challenge, and safety assessments were limited to macroscopic observations. As this study was designed as a proof-of-concept investigation, we focused primarily on evaluating the immunogenicity and mucosal responses elicited by the intranasal formulation. Nevertheless, we observed no overt clinical signs of illness, significant weight loss, or changes in behavior, appetite, or general appearance in immunized mice compared to controls, and no major lung pathology was noted upon macroscopic examination during organ collection. We also acknowledge that the current manuscript does not provide direct experimental evidence demonstrating the efficiency and stability of RBD adsorption onto *B. subtilis* DG101 spores. Although this statement is grounded on previously established adsorption methods found in the literature [52,53,54,55,56] and supported by theoretical fundamentals and the strong immunogenicity observed, we recognize that a detailed characterization of the adsorption process would strengthen the mechanistic understanding of our platform. To further support translational potential, future studies should validate in vivo protection using appropriate challenge models and perform comprehensive safety profiling, including detailed histopathological analyses, stringent monitoring of clinical parameters, and assessment of potential spore germination or replication in the respiratory tract. In addition, further research should explore clinical development of this mucosal vaccine strategy, including its adaptation to multivalent designs or its use in next-generation formulations targeting emerging respiratory pathogens. Nevertheless, it is important to note that both *B. subtilis* spores (particularly the human probiotic DG101 strain) and recombinant RBD protein have well-established safety profiles [116,117], as highlighted in the Introduction and Discussion sections, supporting the theoretical safety of our formulation.

## Figures and Tables

**Figure 2 vaccines-13-00772-f002:**
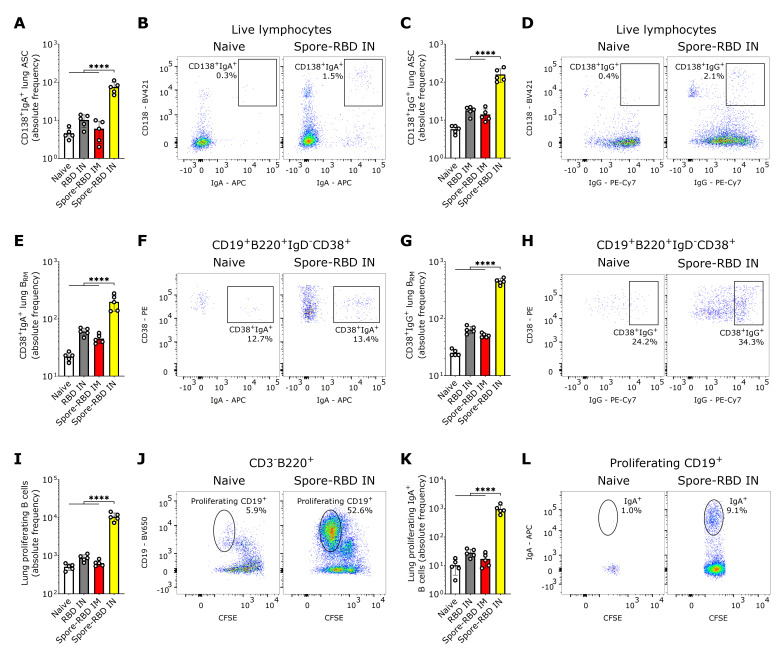
SARS-CoV-2 RBD intranasal immunization adjuvanted with *Bacillus subtilis* DG101 spores induces mucosal B cell memory in the lungs. Spleen and lungs were harvested on day 42. The data is shown in absolute frequency and representative fluorescence bivariate histograms. (**A**–**D**) CD138^+^IgA^+^ASCs (**A**,**B**) and CD138^+^IgG^+^ASCs (**C**,**D**) in the lungs of naive mice, or mice immunized with RBD alone intranasally (RBD IN), or with RBD adsorbed onto *B. subtilis* DG101 spores either intramuscularly (spore-RBD IM) or intranasally (spore-RBD IN). (**E**–**H**) CD38^+^IgA^+^B_RM_ (**E**,**F**) and CD38^+^IgG^+^B_RM_ (**G**,**H**) in the lungs of naive, RBD IN, spore-RBD IM, and spore-RBD IN. (**I**–**L**) Lung proliferating CD19^+^B220^+^ B cells (**I**,**J**) and subset CD19^+^B220^+^IgA^+^ B cells (**K**,**L**), isolated from the lungs of naive, RBD IN, spore-RBD IM, and spore-RBD IN, following in vitro antigen recall in a CFSE proliferation assay. All data are depicted as mean ± SD. Each experimental group consists of five animals, and each dot represents an individual animal. Statistical significance was calculated by one-way analysis of variance (ANOVA); **** *p*  <  0.0001.

**Figure 3 vaccines-13-00772-f003:**
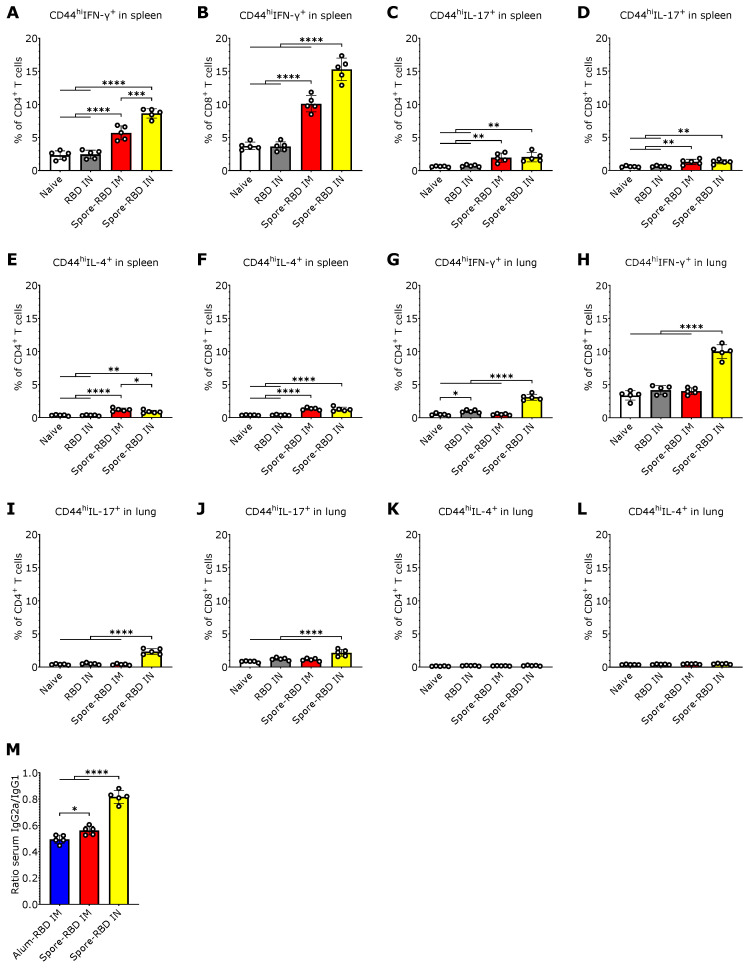
SARS-CoV-2 RBD intranasal immunization adjuvanted with *Bacillus subtilis* DG101 spores induces systemic and mucosal T cell immunity with a Th1 bias. Spleen and lungs were harvested on day 42. (**A**–**L**): Percentage of Th1 (CD44^hi^IFN-γ^+^CD4^+^) from total CD4^+^ T cells (**A**,**G**), Tc1 (CD44^hi^IFN-γ^+^CD8^+^) from total CD8^+^ T cells (**B**,**H**), Th17 (CD44^hi^IL-17^+^CD4^+^) from total CD4^+^ T cells (**C**,**I**), Tc17 (CD44^hi^IL-17^+^CD8^+^) from total CD8^+^ T cells (**D**,**J**), Th2 (CD44^hi^IL-4^+^CD4^+^) from total CD4^+^ T cells (**E**,**K**), and Tc2 (CD44^hi^IL-4^+^CD8^+^) from total CD8^+^ T cells (**F**,**L**) in spleens and lungs from naive mice, or mice immunized with RBD alone intranasally (RBD IN), or with RBD adsorbed onto *B. subtilis* DG101 spores either intramuscularly (spore-RBD IM) or intranasally (spore-RBD IN). (**M**) Specific IgG2a/IgG1 ratio in 42-day serum of BALB/c mice immunized with recombinant SARS-CoV-2 RBD adsorbed onto aluminum hydroxide intramuscularly (alum-RBD IM), or onto *B. subtilis* DG101 spores intramuscularly (spore-RBD IM) or intranasally (spore-RBD IN). All data are depicted as mean ± SD. Each experimental group consists of five animals, and each dot represents an individual animal. Statistical significance was calculated by one-way analysis of variance (ANOVA); * *p*  <  0.05, ** *p*  <  0.01; *** *p*  <  0.001 or **** *p*  <  0.0001.

**Figure 4 vaccines-13-00772-f004:**
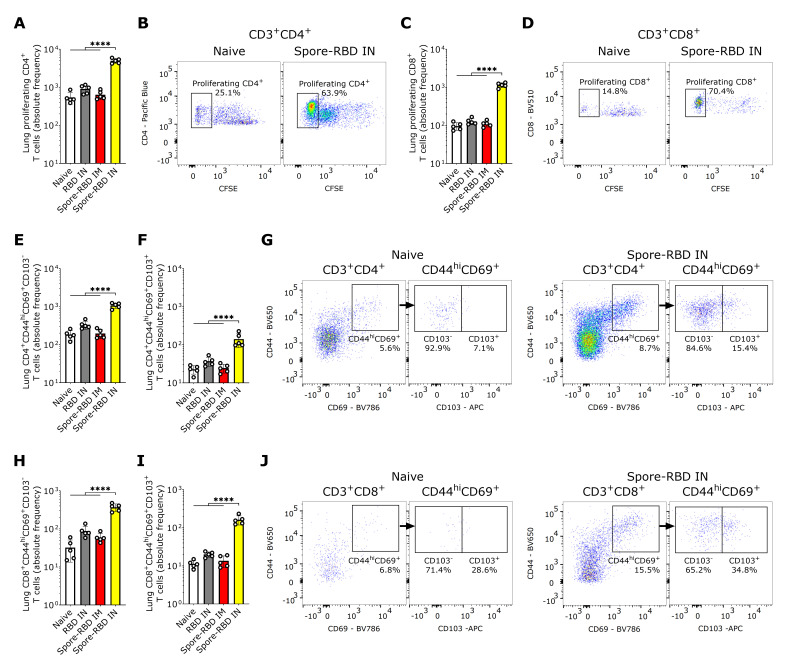
SARS-CoV-2 RBD intranasal immunization adjuvanted with *Bacillus subtilis* DG101 spores induces mucosal T cell memory in the lungs. Spleen and lungs were harvested on day 42. The data is shown in absolute frequency and representative fluorescence bivariate histograms. (**A**–**D**): Proliferating CD4^+^ (**A**,**B**) and CD8^+^ T cells (**C**,**D**) after in vitro antigen recall in a CFSE proliferation assay of lung cells from naive mice, or mice immunized with RBD alone intranasally (RBD IN), or with RBD adsorbed onto *B. subtilis* DG101 spores either intramuscularly (spore-RBD IM) or intranasally (spore-RBD IN). (**E**–**G**): CD4^+^CD44^hi^CD69^+^CD103^−^ (**E**,**G**) and CD4^+^CD44^hi^CD69^+^CD103^+^ T cells (**F**,**G**) in the lungs of naive, RBD IN, spore-RBD IM, and spore-RBD IN. (**H**–**J**): CD8^+^CD44^hi^CD69^+^ CD103^−^ (**H**,**J**) and CD8^+^CD44^hi^CD69^+^CD103^+^ T cells (**I**,**J**) in the lungs of naive, RBD IN, spore-RBD IM, and spore-RBD IN. All data are depicted as mean ± SD. Each experimental group consists of five animals, and each dot represents an individual animal. Statistical significance was calculated by one-way analysis of variance (ANOVA); **** *p*  <  0.0001.

## Data Availability

The data presented in this study are available upon request from the corresponding author.

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
