# Peer review of "Mucosal Vaccination Against SARS-CoV-2 Using Human Probiotic Bacillus subtilis Spores as an Adjuvant Induces Potent Systemic and Mucosal Immunity"

_vaccines, 2025, doi:10.3390/vaccines13070772_

Round 1

Reviewer 1 Report

Comments and Suggestions for Authors

In this study a novel mucosal vaccination strategy against SARS-CoV-2, consisting of RBD protein formulated with Bacillus subtilis spores, is evaluated in mice for immunogenicity. A comparison is made with intramuscular vaccination strategies, using the same immunogen with either the same adjuvant or formulated in alum. The study is well performed and provides robust data to support the superior systemic and mucosal immunogenicity provided by intranasal application of RBD protein with Bacillus subtilis spores. The results are interesting, the study is well performed and the paper is well written.  

Comments

  1. In the abstract it is claimed that the immunization strategy supports “effective anti-viral immunity and long-term protection”. However, efficacy is not evaluated in this study and this conclusion has to be modified.
  2. Since the use of Bacillus subtilis spores is not well known to most readers some additional background information about its application and possible mechanism of action should be provided in the introduction.
  3. From the materials and methods section it is not entirely clear how the T cell responses were measured. Were the cells stimulated with RBD protein as described for the CFSE-proliferation assay? Were golgiplug or golgistop used during the assay to induce accumulation of cytokines inside the cell?
  4. The results section, line 331-334 is not very clear. From the figures it appears that the intranasal immunization results in relatively higher levels of IgG2a subclass Ab relative to IgG1, which is indicative for a more Th1 biased response. Is that correct? From figure 3H it can be inferred that also in the intranasal vaccine group the IgG2a/IgG1 ratio is still below 1. Therefore, there is still a slight Th2 bias. This should also be clear from the conclusion given in the results section. Could this sentence be rephrased?
  5. In figure 1 the lines indicating significant differences between the groups are hardly visible. In this and the other figures the letters are quite small.
  6. In figure 2 and 4 the legend mentions “frequency”, while in the graphs “total” is mentioned. Are the total number of cells falling into the gate displayed or is it the frequency. For instance, in figure 2E-H there is indeed a clear increase in CD38+IgA+ and CD38+IgG+ cells. However, also the CD38+IgA- and CD38+IgG- populations seem to be increased. Is there an influx of all B cells in the immunized animals, leading to an increase in IgA and IgG memory B cells? Or is there really an increase in percentage?
  7. In the results section it is described that for figure 2 I-L a CD3-CD19+B220+ gating was used. However, in the graphs also a population of CD19 negative cells can be seen. Also here it is not clear whether total cell number or percentage of proliferating B cells is shown.
  8. In Figure 4 E-F and H-I both the population of CD103- and CD103+ T cells is highest in the intranasally immunized animals. In the results section it is said that Trm are expanded. However, is this then not equally the case for the non-Trm? To draw this conclusion about Trm expansion the ratio between CD103+ and CD103- should also increase.

Reviewer 2 Report

Comments and Suggestions for Authors

In the manuscript, Pupo et al. evaluated an intranasal vaccine regimen comprising recombinant RBD adjuvanted with Bacillus subtilis DG101 spores, a human probiotic, in mice. The study evaluated systemic and mucosal immune responses post-vaccination, suggesting that intranasal RBD vaccines adjuvanted with B. subtilis spores may induce sterilizing immunity against SARS-CoV-2. However, several concerns warrant further consideration:

  1. The intranasal dose (80 µg/mouse) used in this study is substantially higher than those reported in prior studies. Please provide a rationale for this selection.
  2. Surprisingly, elevated IgM levels were still detectable at Day 35/42 post-booster (Fig. 1C), which is unusual for this timepoint. Could the authors elaborate on this observation?
  3. The manuscript mentions that "additional serum was collected on Day 180 to evaluate long-term immunity," but these data are not presented. Please include the results.
  4. The IN group showed stronger systemic antibody responses than the IM group (Fig. 1), which contrasts with findings from other studies.
  5. As a mucosal vaccine, the study should include neutralizing antibody titers from nasal wash and BALF to fully assess mucosal immunity.
  6. The IN group exhibited a higher IgG2a/IgG1 ratio than the IM group (Fig. 3M). What could explain this higher Th1-skewed response in the intranasal regimen?
  7. All flow cytometry figures should include the frequency (%) of relevant cell populations to facilitate data interpretation.
  8. The potential advantages of Bacillus subtilisDG101 spores as a novel mucosal adjuvant should be more thoroughly discussed, particularly in comparison to established adjuvants.
  9. The discussion should explicitly address how DG101 spores outperform or differ from other mucosal adjuvants in terms of efficacy, safety, or immune modulation.

Reviewer 3 Report

Comments and Suggestions for Authors

Summary: Authors report a detailed systemic and mucosal immune response profile to an experimental intranasal SARS-CoV2 vaccine based upon purified RBD protein absorbed to Bacillus subtilis DG101 spores. Mice were vaccinated intranasally on days 0, 14, and 28 and compared to intramuscular vaccine, spores alone, alum adjuvant and RBD protein alone. Antibody assays and immunophenotyping results include: 1) systemic and humoral responses (serum, saliva, BALF, and NALT antibody responses and lung RDB+ B cells), 2) B-cell memory responses in the lung (ASC IgG and IgA, BRM IgG and IgA), 3) IFN-g, IL-4 and IL17 T-cell responses in spleen and lung, and 4) T cell memory responses in the lungs.  Authors conclude that the vaccine induces significant antigen-specific mucosal and systemic humoral responses, B and T cell memory and a skewed Th1/Tc1 T cell response.

This report is deemed important and interesting. Results clearly show that the vaccine induces significant B and T cell immune responses- importantly including extensive analysis of mucosal and memory responses, which are often overlooked. 

Main comments:

Methods: Does the RBD protein effectively absorb to the surface of the spores? Evidence of this, or previous reports for references, should be included to support mechanism of action.

Methods/Results: Authors indicate that there was no evidence of inflammation in the respiratory tract but do not show results. Were there any clinical assessments done on the mice as evidence of safety -such as temperature changes and weight loss? Was there evidence of Bacillus spore germination/replication? Please include. 

ELISAs (Figure 1): only one graph in this figure reported titers to calculate relative antibody responses  (serum anti IgG, Figure 1E). If titers were not determined for all other graphs/samples, then need to report what dilution of sample (serum, saliva, BALF, NALT supernatant) was used for OD450 determination.  

ELISAs (Figure 1): it is not reported what day of the trial these samples were taken for ELISA – sometime between 35-42 or day 180?

General figures: Statistical significance is not shown for all groups analyzed in ANOVA in most graphs. For example, in Figure 1B – what is p-value between spore-RBD IM and spore-RBD IN?

Figures 2-4: A table of antibodies and/or improved figure labelling is needed to better explain immunophenotyping. Figure 2 (CD138, B220, CD38) is especially confusing and not well described in the text, with readers expected to know all marker significance.

Conclusions: authors suggest a skewed Th1/Tc1 response “without Th2 or Tc2 expansion” but results indicate a strong B cell and antibody response as well as a IgG2a/IgG1 ratio of 0.8. There is low IL4 and high INF-g response, but this was true for all groups tested. Please rewrite/explain conclusion of Th1/Tc1 bias in more detail.

Minor comments:

Graphs are generally small and font size too small on labels to read

Round 2

Reviewer 2 Report

Comments and Suggestions for Authors

Thank you for the responses. However, some minor revisions are still required. For instance, there are errors in Figure 1E that need correction, and the corresponding figure legend should include the sample collection time. For clarity, it would be preferable to replace the abbreviations 'in' and 'im' with either uppercase ('IN,' 'IM') or standard subscript notation ('i.n.,' 'i.m.') in both the figures and main text. Thorough proofreading to polish the writing is needed.

Author Response

Comments 1: Thank you for the responses. However, some minor revisions are still required. For instance, there are errors in Figure 1E that need correction, and the corresponding figure legend should include the sample collection time. For clarity, it would be preferable to replace the abbreviations 'in' and 'im' with either uppercase ('IN,' 'IM') or standard subscript notation ('i.n.,' 'i.m.') in both the figures and main text. Thorough proofreading to polish the writing is needed.

Response 1: We thank the reviewer for the valuable comments and careful reading. We have corrected Figure 1E and updated the legend in lines 388 and 389 to more explicitly include the sample collection time. This has also been done for all the figures. To improve readability and ensure uniformity and consistency, we have replaced all abbreviations ‘in’ and ‘im’ with the uppercase 'IN' and 'IM' notation throughout the main text and all the figures. Additionally, we have corrected the use of decimal points (replacing commas with periods) where needed and ensured appropriate notation for cytokines (i.e., IL-4, IL-17, IFN-γ) and other minor corrections throughout the text and figures. Finally, we have performed detailed proofreading and revised the manuscript to enhance clarity and polish the writing. For this commitment, we gained support from Grammarly, an advanced writing assistant software tool, to further improve the grammar, style, and overall readability. We appreciate the reviewer's attention to these constructive suggestions, which have significantly enhanced the manuscript.